# Opsonic Activity of Conservative Versus Variable Regions of the Group A *Streptococcus* M Protein

**DOI:** 10.3390/vaccines8020210

**Published:** 2020-05-07

**Authors:** Chuankai Dai, Zeinab G. Khalil, Waleed M. Hussein, Jieru Yang, Xiumin Wang, Lili Zhao, Robert J. Capon, Istvan Toth, Rachel J. Stephenson

**Affiliations:** 1School of Chemistry and Molecular Biosciences, The University of Queensland, St. Lucia, QLD 4072, Australia; chuankai.dai@uq.net.au (C.D.); w.hussein@uq.edu.au (W.M.H.); jieru.yang@uq.edu.au (J.Y.); wangxiumin@caas.cn (X.W.); lili.zhao@uq.net.au (L.Z.); i.toth@uq.edu.au (I.T.); 2Institute for Molecular Bioscience, The University of Queensland, St. Lucia, QLD 4072, Australia; z.khalil@uq.edu.au (Z.G.K.); r.capon@imb.uq.edu.au (R.J.C.); 3Pharmaceutical Organic Chemistry Department, Faculty of Pharmacy, Helwan University, Helwan 11795, Egypt; 4Gene Engineering Laboratory, Feed Research Institute, Chinese Academy of Agricultural Sciences, Beijing 100081, China; 5Key Laboratory of Feed Biotechnology, Ministry of Agriculture and Rural Affairs, Beijing 100081, China; 6School of Pharmacy, The University of Queensland, Woolloongabba, QLD 4102, Australia

**Keywords:** opsonization, M protein, J8-epitope, 88/30-epitope, Group A *Streptococcus*, peptide-based subunit vaccine

## Abstract

Group A *Streptococcus* (GAS) and GAS-associated infections are a global challenge, with no licensed GAS vaccine on the market. The GAS M protein is a critical virulence factor in the fight against GAS infection, and it has been a primary target for GAS vaccine development. Measuring functional opsonic antibodies against GAS is an important component in the clinical development path for effective vaccines. In this study, we compared the opsonic activity of two synthetic, self-adjuvanting subunit vaccines containing either the J8- or 88/30-epitope in Swiss outbred mice using intranasal administration. Following primary immunization and three boosts, sera were assessed for IgG activity using ELISA, and opsonization activity against seven randomly selected clinical isolates of GAS was measured. Vaccine constructs containing the conservative J8-epitope showed significant opsonic activity against six out of the seven GAS clinical isolates, while the vaccine containing the variable 88/30-epitope did not show any significant opsonic activity.

## 1. Introduction

Group A *Streptococci* (GAS) are Gram-positive bacteria responsible for many infections and diseases. GAS infections range from uncomplicated pharyngitis, cellulitis and pyoderma to life-threatening infections that include *necrotizing fasciitis*, streptococcal toxic shock syndrome, *sepsis* and *pneumonia*. GAS-infections also cause serious autoimmune diseases, the most significant being acute rheumatic fever and rheumatic heart disease [1]. Worldwide, there are an estimated 600 million people affected with GAS infection each year, with GAS disproportionally affecting disadvantaged populations [2]. In 2013, a Global Burden of Disease study estimated that rheumatic heart disease affected 34,000,000 people worldwide with more than 345,000 deaths per year [3]. Further, rheumatic heart disease has contributed to disease burdens, totaling over 10,000,000 disability-adjusted life years. The mortality rates associated with rheumatic heart disease and the invasive deaths associated with GAS infections (an additional 160,000 deaths per year) put GAS as the fifth leading cause of single-pathogen infectious disease deaths, behind human immunodeficiency virus (HIV), tuberculosis, malaria, and *Streptococcus pneumonia* [3]. Currently, antibiotics (e.g., penicillin) are the primary treatment for GAS infection, but antibiotic resistance is becoming a concern [4]. A vaccine to address the global burden of GAS would reduce the rates of GAS-associated infections and deaths, but to date, a safe and effective commercial vaccine is currently not available [5]. 

Peptides as antigens are a modern vaccine approach that uses minimal microbial components to stimulate adaptive immunity against a pathogen [6]. Peptides are seen as a safer alternative to using the whole organism or protein, which in the case of GAS, have been associated with allergic and autoimmune responses [6]. The GAS M protein (Figure 1), a coiled-coil homodimer surface-anchored protein encoded by the *emm* gene, has been identified as one of the major virulence factors of GAS infection preventing opsonophagocytosis, and as a result, has been a major focus in GAS vaccine development [7]. However, due to the cross-reactivity of the M protein with human cardiac cells, peptide antigens derived from the M protein have the potential to provide protection against a broad spectrum of GAS strains while free from any autoimmune responses. More specifically, the J8i minimal B cell epitope (**SREAKKQVEKAL**) has been identified from the C repeat region of the M protein and is recognized by human sera antibodies of most living adults in GAS endemic areas. This J8i peptide sequence was also capable of stimulating humoral immunity in vivo [8]. Flanking the J8i peptide with the GCN4 DNA binding protein sequence produced the J8-epitope (QAEDKVKQ**SREAKKQVEKAL**KQLEDKVQ), which has been shown to maintain the M protein epitopes native α-helical confirmation [8,9]. A peptide vaccine containing the J8-epitope (adjuvanted with Alum or Saponin-based adjuvants-2) has triggered the production of opsonic immunoglobulin G (IgG) antibodies in mice, providing protection against a systemic challenge [10,11]. It was recently reported that the J8-epitope covers 37% of the 2083 isolates and J8’s variants, J8.12 and J8.40, covering 79% and 76% of 2083 GAS genomes, respectively. This suggested that vaccines containing the J8-epitope would be highly broadly protective, with proof of this being the recent clinical evaluation for the J8 peptide vaccine (adjuvanted with diphtheria toxoid) (MJ8VAX) [12,13]. Additionally, Hayman et al. reported that a J8 peptide vaccine (adjuvanted with Complete Freund’s adjuvant) generated high antibody production (titer >12,800) in inbred mice following primary immunization and four boosts. However, these antibodies only opsonized 49% of the GAS bacteria, with speculation that the antibody recognition site on the GAS bacteria tested in the opsonization assay was hindered by the presence of the hyaluronic acid capsule, reducing antibody binding and cell death [14,15]. From this, a GAS vaccine containing epitopes outside the GAS M protein C-terminal region would assist with antibody binding and increased opsonization activity.

The 88/30-epitope (DNGKAIYERARERALQELGP), also known as 88/30_1-20_, was identified from the *N*-terminal region of the M protein on GAS skin 88/30 isolate (*emm*97, Figure 1) [17,18,19]. The 88/30-epitope has shown a high opsonization capacity (>90%) to its original GAS 88/30 bacteria, with the ability to cross-react with other GAS isolates, including GAS 2034 and GAS Y530S (opsonization capacity >90%) [20]. Olive et al. proved that in a murine model, peptide vaccines containing the 88/30-epitope were able to enhance a significant induction of antibodies, providing excellent protection to the challenged mice against selected GAS isolates without any additional adjuvant [19]. Further, McNeil et al. and Kotloff et al. suggested that a recombinant multivalent protein vaccine candidate based on *N*-terminal GAS antigens elicited high opsonic activities, enabling opsonization of the corresponding GAS bacteria [21,22]. Vaccine safety in the clinical trials has also indicated that there is no cross-reactivity between the antibody and human tissues [21,22]. On top of this, Moyle et al. reported that a multi-epitope polypeptide synthetic vaccine containing both the 88/30- and J8-epitopes was able to elicit high sera antibody production with the 88/30-epitope generating higher IgG compared with the J8-epitope [23]. Although *N*-terminal epitope vaccines (e.g., the 6-valent and 25-valent protein vaccine developed from high variable regions of M protein) have shown preclinical cross-protection against other strains of GAS, cross-protection has been limited within the same *emm*-clusters [15]. From this, vaccines based on *N*-terminal region epitopes need to be carefully designed to cover the targeting strains. An example of this is an animal trial for a GAS *N*-terminal 30-valent vaccine (StreptAnova^®^, consisting of four fusion proteins comprised of high the variable region peptide from 30 M-serotypes and adjuvanted with alum), developed with 98% protection in the United States and 78% in Europe when tested in rabbits. The vaccine also protected 43 of 53 non-vaccine serotypes with an 80% opsonization rate [15,24]. Currently, the StreptAnova^®^ vaccine has completed phase I clinical trials in humans, with the results showing low autoimmunity and effectivity eliciting antibodies against 24 of the 30 M protein antigens. Further to this, a 66% mean specific opsonic rate against 30 *emms* was also identified [3,25]. As the StreptAnova^®^ vaccine is developed from GAS strains isolated in the United States and Europe, with worldwide variation in GAS isolates, the development of a multivalent GAS vaccine remains a challenge [24]. For example, the 88/30-epitope is only significantly aligned in 34 of the 2149 GAS reported sequences (1.5%) from the US CDC Blast–*emm* and *emm* databases (searched on 27 February 2020), which suggested that the 88/30-epitope was not broadly protective among reported clinical isolates globally. Interestingly, the 88/30-epitope was one of the clinical isolates that made up 13% of the local community isolates from the Royal Darwin Hospital, Northern Territory, Australia, and hence, is important in the development of GAS vaccines for the Australian population [20]. 

In research, an in vitro opsonization assay is used to determine the sera opsonic activity of an IgG antibody (Figure 2A). Measuring functional opsonic antibodies is an important component in the clinical development path for vaccine development. Here, opsonization refers to an immune process where particles, such as bacteria, are targeted for destruction by an immune cell, known as antibody-mediated phagocytosis. The process of opsonization is a means of identifying the invading particle to the phagocyte, leading to killing of the bacterial cell. Once the antigen-specific IgG antibody binds to the corresponding antigen, the CH2 domain on the Fc region of the IgG antibody are able to bind to the IgG Fc receptor (FcR) on the macrophage to enhance the engulfing and digestion of bacteria cells, which are marked by the IgG antibody (Figure 2B) [26]. 

In real world clinical analysis for vaccines, GAS infections occur randomly and with constant mutation of the GAS genome. However, there was no direct comparison of the opsonic activity of the C-terminus (J8) and N-terminus (88/30) epitopes in our vaccine delivery system. In this study, to evaluate the opsonic activities of the J8- and 88/30-epitopes used in vaccine development, antibodies were produced by synthetic peptide-based vaccine constructs following intranasal administration to Swiss outbred mice. Antibodies were tested for their ability to opsonize seven random clinical isolates of GAS to compare the vaccines opsonic ability. Given that all clinical isolates of GAS used in this study originated in Australian hospitals, we assumed that the broader protecting J8-epitope (J8 has been shown to cover 37% of the 2083 isolates and J8’s variants, J8.12 and J8.40, cover 79% and 76% of 2083 GAS genomes, respectively) would provide better protection compared to the 88/30-epitope (1.5% of GAS isolates for GAS and 13% of the local community isolates from the Royal Darwin Hospital, Northern Territory, Australia), as detailed previously [12,17]. 

## 2. Materials and Methods

### 2.1. General Methods

All chemicals used were of analytical grade unless otherwise stated. Resin and amino acids were purchased from Novabiochem (Läufelfingen, Switzerland) and Mimotopes (Melbourne, VIC, Australia). Peptide synthesis grade dichloromethane (DCM), *N,N*-dimethylformide (DMF), 1-[Bis(dimethylamino)methylene]-1H-1,2,3-triazolo[4,5-b]pyridinium 3-oxid hexafluorophosphate (HATU), hydroxybenzotriazole (HOBt), diethyl ether, high performance liquid chromatography (HPLC) grade methanol (MeOH), acetonitrile (MeCN) and other solvents were purchased from Merck (Kilsyth, VIC, Australia) or Mimotopes (Melbourne, VIC, Australia). Peptide synthesis grade trifluoroacetic acid (TFA) was purchased from Novachem (Heidelberg West, VIC, Australia). All other reagents were purchased from Merck (Kilsyth, VIC, Australia) or Sigma-Aldrich (Castle Hill, NSW, Australia). Diisopropylethylamine (DIPEA) was purchased from Fujifilm Wako Pure Chemicals (Wako, Saitama, Japan). Goat anti-mouse IgG (H+L)-HRP conjugate secondary antibody was purchased from Bio-Rad (Gladesville, NSW, Australia). SIGMA*FAST*^TM^
*O*-phenylenediamine dihydrochloride (OPD) tablets were purchased from Sigma-Aldrich (Castle Hill, NSW, Australia): each group of two tablets (one gold and one silver) was dissolved in 20 mL Milli-Q water, containing 0.4 mg/mL OPD, 0.4 mg/mL urea hydrogen peroxide and 0.05 M phosphate citrate.

Microwave-assisted peptide synthesis was performed assisted by the CEM Discovery microwave peptide synthesizer (CEM Corp, Matthews, NC, USA). Analytical RP-HPLC was performed on a Shimadzu (Kyoto, Japan) liquid chromatography system (SPD-M10A VP, LC-20AB, DGU-20A5, SIL-20AC HT). Samples were analyzed using a Grace Vydac 218TP C18 column (5 μm, 4.6 mm diameter, 50 mm length) or Vydac 214TP C4 column (5 μm, 3.6 mm diameter, 250 mm length), with an 1 mL/min flow rate with solvent A (Milli-Q water with 0.1% TFA) and solvent B (90% MeCN in Milli-Q water with 0.1% TFA). Preparative reverse phase (RP)-HPLC was performed on a Shimadzu liquid chromatography system (SPD-20A, LC-20AP, CBM-20A, FRC-10A). Samples were purified using the Grace Vydac 214TP1022 Protein C4 column or Vydac 218TP1022 Protein & Peptide C18 column, with a 20 mL/min flow rate. All RP-HPLC data were collected and analyzed with LabSolutions 5.51 software. Electrospray ionization mass spectrometry (ESI-MS) was performed on the PE Sciex API3000 triple quadrupole mass spectrometer (Concord, ON, Canada), and MS data were collected and analyzed with Analyst 1.4 software. Dynamic light scattering (DLS) measurements were taken using a Zetasizer Nano ZP instrument (Malvern Instrument, Malvern, UK) with Dispersion Technology software. Particle photograph were captured using a JEM-1010 transmission electron microscope (TEM) (JEOL Ltd., Tokyo, Japan). 96-well microtest plates (flat base, polystyrene, high binding) were purchased from Sarstedt (Mawson Lakes, SA, Australia). ELISA Optical Density (O.D.) values were read by the SpectraMAX 250: 96-well plate reader purchased from Molecular Devices (San Jose, CA, USA). All data were analyzed with GraphPad Prism 8.3.1 software.

### 2.2. Vaccine Synthesis

**VC-1** and **VC-2** (structures shown in Figure 3) were synthesized using microwave-assisted solid phase peptide synthesis (SPPS) using Boc chemistry on *p*MBHA HCl resin (0.59 g/mmol loading) following methods previously reported [30]. Boc-protected amino acids (4 eq.) were activated with HATU (0.5 M solution in DMF, 4 eq.) and DIPEA (6.2 eq.) for 5 min before coupling (1 × 5 min; 1 × 10 min; 70 °C, 20 W). Boc deprotection was achieved using neat TFA (2 × 1 min with stirring). Fmoc (on lysine side chain) was used for orthogonal branching for addition of the GAS epitopes (J8 or 88/30). Fmoc was deprotected with 20% piperidine in DMF solution (1 × 5 min; 1 × 10 min; 70 °C; 20 W). Acetylation of the peptide N-terminal (1 × 5 min; 1 × 10 min; 70 °C; 20 W) was achieved with freshly prepared acetic anhydride solution (0.5 mL acetic anhydride, 0.5 mL DIPEA, 5 mL DMF). Peptide cleavage was achieved with hydrofluoric acid (10 mL per 1 gram resin, with 5% *p*-cresol and 5% *p*-thiocresol as scavengers) at 8 °C. Crude lipo-peptides were washed with diethyl ether:hexane (1:1) solution, dissolved with solvent A:B (1:1) and lyophilized.

**VC-3** (structure shown in Figure 3) was synthesized using microwave-assisted SPPS via Fmoc chemistry on Rink amide MBHA resin (0.52 g/mmol loading) following methods previously reported [31]. Fmoc-protected amino acids (4 eq.) were activated with HATU (0.5 M solution in DMF, 4 eq.) and DIPEA (5.2 eq.) for 5 min before coupling (1 × 5 min; 1 × 10 min; 70 °C, 20 W). Fmoc deprotection was achieved using 20% piperidine in DMF solution (2 × 2 min; 70 °C, 20 W). After the coupling, Fmoc-Asp(OtBu)-OH, the deprotection was changed into 20% piperidine (0.1 M HOBt, DMF solution, 2 × 5 min; 50 °C, 20 W). Protecting group ivDde (on lysine side chain) was used for orthogonal branching for addition of the GAS epitopes (J8). ivDde was deprotected with 2% hydrazine in DMF solution (30 × 15 min; R.T. with stirring). Acetylation of the N-terminal (1 × 5 min; 1 × 10 min; 70 °C; 20 W) was achieved with freshly prepared acetic anhydride solution (0.5 mL acetic anhydride, 0.5 mL DIPEA, 5 mL DMF). Peptide cleavage was achieved with TFA (10 mL per 1 gram resin, with 2.5% water, 2.5% 1,2-ethanedithol and 1% tri-isopropylsilane) at R.T. Crude **VC-3** peptide was washed with diethyl ether, dissolved with solvent A:B (1:1) and lyophilized.

#### 2.2.1. **VC-1**

**VC-1** (0.15 mmol scale) was purified using preparative RP-HPLC on C4 column with a solvent gradient of 35–65% solvent B over 45 min, and checked with analytical RP-HPLC on C4 column: *T_R_* = 25.0 min, purity >95%. MW: 6014.2 Da.; ESI-MS: [M+4H]^4+^ = 1503.9 (calculated 1504.6); [M+5H]^5+^ = 1203.9 (calculated 1203.8); [M+6H]^6+^ = 1004.0 (calculated 1003.4); [M+7H]^7+^ = 860.5 (calculated 860.2); [M+8H]^8+^ = 752.7 (calculated 752.8) (see Appendix A).

#### 2.2.2. **VC-2**

**VC-2** (0.05 mmol scale) was purified using preparative RP-HPLC on C4 column with a solvent gradient of 35–65% solvent B over 45 min, and checked with analytical RP-HPLC on C4 column: *T_R_* = 26.1 min, purity >95%. MW: 5018.0 Da.; ESI-MS: [M+3H]^3+^ = 1673.5 (calculated 1673.7); [M+4H]^4+^ = 1255.4 (calculated 1255.5); [M+5H]^5+^ = 1004.3 (calculated 1004.6); [M+6H]^6+^ = 837.4 (calculated 837.3) (see Appendix A).

#### 2.2.3. **VC-3**

**VC-3** (0.1 mmol scale) was purified using preparative RP-HPLC on C18 column with a solvent gradient of 30–55% solvent B over 45 min, and checked with analytical RP-HPLC on C18 column: *T_R_* = 16.5 min, purity >95%. MW: 5507.4 Da.; ESI-MS: [M+4H]4^+^ = 1377.7 (calculated 1377.9); [M+5H]^5+^ = 1102.2 (calculated 1102.5); [M+6H]^6+^ = 918.8 (calculated 918.9); [M+7H]^7+^ = 787.5 (calculated 787.8); [M+8H]^8+^ = 689.1 (calculated 689.3) (see Appendix A).

### 2.3. Dynamic Light Scattering

**VC-1** and **VC-2** were dissolved in Milli-Q water and analyzed using DLS to measure the particle size and polydispersity index (PDI). All samples were diluted 10 times with Milli-Q water, transferred to disposable cuvettes before measurement and the measurements were taken at 25 °C and 173° light scattering [30,32]. DLS spectra are shown in the Appendix A.

### 2.4. Transmisssion Electron Microsocpy

**VC-1** and **VC-2** were dissolved in Milli-Q water and applied to glow-discharged carbon-coated copper 200 mesh grids and negative-strained with 1% uranyl acetate. The photographs captured were operated at 80 kV [30,32]. TEM images are shown in the Appendix A.

### 2.5. In Vivo Immunogenicity Study

All animal protocols were approved by the university ethics committee (Animal Ethics Unit, Office of Research Ethics, University of Queensland; approval number: SCMB/AIBN/069/17) in accordance with the National Health and Medical Research Council (NHMRC) of Australia. ARC Swiss outbred mice (female, 6–7 weeks old, purchased from the Animal Resource Centre, Perth, WA, Australia) were housed in the Australian Institute for Bioengineering and Nanotechnology (AIBN) Animal Facility. Mice were acclimatized for 7 days before starting the experiment. On day 0, mice were anesthetized with isoflurane gas before being intranasally administered with 60 μg vaccine construct (**VC-1** or **VC-2**) dissolved in 20 μL sterile Milli-Q water (10 μL/nostril). Mice that received the positive control were intranasally administered 60 μg **VC-3** adjuvanted with 10 μg CTB dissolved in 20 μL sterile Milli-Q water (10 μL/nostril). The group that received the negative control were intranasally administered 20 μL sterile Milli-Q water only. All mice received three boosts on days 21, 42 and 63, following the same procedure outlined above.

Blood was collected on days 0, 20, 41, 63 via tail bleed and day 77 via cardiac bleed, and centrifuged (2000 G, 20 min) to collect supernatant sera. Samples were stored at −80 °C.

### 2.6. ELISA Study

The enzyme-linked immunosorbent assay (ELISA) was performed to measure J8- and 88/30-specific IgG antibodies [30]. Plates were coated with a solution of J8 peptide (or 88/30 peptide) in carbonate coating buffer (pH = 9.6, 100 µL/well) and incubated at 37 °C for 90 min. The epitope coated plates were then treated with a blocking solution (150 µL/well) made of 5% skim milk in PBS with 0.05% tween-20 to reduce the non-specific binding. These plates were incubated overnight at 4 °C followed by washing with distilled water (3 times) and PBS with 0.05% tween-20 (3 times). The 100 times diluted individual mouse serum was added to the coated-blocked plates (100 µL/well) and serially diluted (2 times) followed by incubation at 37 °C for 90 min. To detect the epitope-specific antibodies of mice, horseradish peroxidase (HRP)-conjugated goat anti-mouse IgG (H + L) antibody with a 1:3000 in 0.5% skim milk containing 0.05% tween-20 was added to the plates and incubated for 90 min at 37 °C. Following washing, OPD solution (100 µL/well) was added to the plates in the dark as an HRP substrate. Reaction between OPD and HRP was stopped by the addition of 1N H_2_SO_4_ to the plates after incubation in the dark at R.T. for 15 min and measured the absorbance at 450 nm. Antibody titers are defined as the lowest dilution with an optical density more than three times the standard deviation greater than the average value of the optical density of the sera collected from naïve mice. Statistical significances were calculated with a one-way ANOVA followed by Tukey’s post hoc test with GraphPad Prism software.

### 2.7. In Vitro Opsonisation Study

An opsonization assay was performed using a method that has been previously published [12,32,33]. The bactericidal activity of immune sera IgG was tested against seven GAS clinical isolates including: D3840 (nasopharynx swabs); GC2 203 (wound swab); ACM-2727 (Royal Brisbane Hospital); ACM-2002 (Royal Brisbane Hospital, human abscess—lymph gland); D2612 (nasopharynx swabs); ACM-5203 (ATCC 19615, pharynx of child followed by an episode of sore throat); and ACM-5199 (ATCC 12344, NCIB 11841, scarlet fever). Bacteria were prepared by streaking on Todd-Hewitt broth (THB) agar plates with 5% yeast extract, then incubated at 37 °C for 24 h. A single colony from the bacterium was transferred to THB (5 mL) with 5% yeast extracts and incubated at 37 °C for 24 h to get the approximately 10^7^ colony forming units (CFU)/mL. The culture was serially diluted 100 times in PBS and an aliquot (10 μL) was mixed with heat-inactivated (50 °C, 10 min) sera (10 μL) collected on day 77 and horse blood (80 μL). The assay was performed in duplicate on three cultures. Bacteria were incubated in a 96-well plate with sera, before 10 μL aliquot was analyzed based on the CFU enumerated from the plates. The opsonic activity was calculated as the percentage reduction in CFU when compared to the negative control (the mice received Mill-Q water) wells, as previously published. Opsonic activity was quantified using the following (Equation (1)).
(1)$$\left(1-\frac{CFU\;in\;the\;presence\;of\;serum}{Mean\;CFU\;in\;the\;presence\;of\;media}\right)\times 100\%$$

Equation (1). Quantification of opsonic activity where CFU is colony forming units [32].

## 3. Results and Discussion 

### 3.1. Vaccine Design and Synthesis 

Vaccines have many routes of administration, including subcutaneous, intramuscular injections, intranasal or oral delivery. Intranasal delivery has many benefits, including decreased pain upon administration and increased patient compliance [34]. However, peptide-based mucosal vaccines require (1) effective delivery systems to preserve the peptide epitope from degradation, and (2) an adjuvant to induce both protective mucosal and systemic immune responses [35,36]. In this study, the lipid-core peptide (LCP) delivery system (S-S-C_16_-C_16_, Figure 3) was incorporated into the vaccine constructs **VC-1** and **VC-2** (Figure 3) to induce self-adjuvanting activity, as it has been shown to target the vaccine to the TLR2 on antigen presenting cells. Previous research suggested that lipids derived from bacterial cell wall components are toll-like receptor 2 (TLR2) targeting moieties [37]. TLR2 is critical in an immune response and is linked with pathogen-associated molecular patterns, inducing the CD4^+^ T helper cell response and an innate immune response [37]. Tripalmitoyl-*S*-glyceryl cysteine (Pam_3_Cys) and dipalmitoyl-S-glyceryl cysteine (Pam_2_Cys), identified from Braun’s lipoprotein from *Escherichia coli*, was developed as a fully-synthetic vaccine adjuvant and applied in the development of vaccines, including the HIV vaccine (in clinical trials) and the GAS vaccine (preclinical trials). Pam_3_Cys has also been approved by the Food and Drug Administration of the United States for the *Neisseria meningitides* vaccine [6,38]. The lipid core peptide is a fully synthetic lipo-peptide-based vaccine delivery system based on lipoamino acids with a long alkyl side chain (e.g., 12 to 20 carbons in length) coupled to a peptide epitope. Lipoamino acids, as self-adjuvanting moieties, have been widely assessed in the preclinical development of vaccine candidates for diseases, including GAS, hookworm, human papillomavirus and *Chlamydia trachomatis* [39,40,41]. Vaccines containing lipid adjuvants have been shown to stabilize peptide antigens from degradation and increase peptide stability during storage [42]. For the purpose of comparison, in **VC-3**, the commercial adjuvant CTB was used. 

The T helper epitope P25 (Figure 3) was included in all vaccine constructs to elicit the CD4^+^ response by targeting the CD40 receptor on T helper cells. T helper epitope is a family of epitopes that target major histocompatibility complex class II (MHC-II) molecules on antigen presenting cells to the CD40 ligand on T helper cells, supporting increased MHC binding, essential for the naïve B cell activation [43]. Previous reports have found that inclusion of a T helper peptide into a synthetic peptide-based GAS vaccine is essential due to the B cell epitopes identified from GAS (e.g., J8) not containing a T cell epitope [8]. The P25 epitope is a T helper epitope derived from the fusion protein of canine distemper virus [44]. P25 has been shown to be essential for recognizing and binding to the peripheral blood mononuclear cells (PBMC), especially CD4^+^ T helper cells, which are critical in the activation of B cells necessary for antibody production. Torresi et al. found that the conjugation of the P25 T helper epitope to B cell epitopes induced a stronger antibody response when compared to the B cell antigen alone in a hepatitis C virus vaccine model in inbred mice [45]. 

To compare the J8- and 88/30-epitopes effectiveness in the opsonization of clinical GAS isolates, a small library of vaccine constructs (Figure 3) containing different GAS M protein epitopes (J8 or 88/30), a T helper epitope (P25) and a synthetic lipid adjuvant (S-S-C_16_-C_16_ for **VC-1** and **VC-2**) or CTB (for **VC-3**) were synthesized to induce the mice eliciting immune responses to production of the antibodies (IgG etc.). **VC-1** and **VC-2** were synthesized using in situ Boc-SPPS on *p*MBHA HCl resin, and **VC-3** was synthesized using in situ Fmoc-SPPS on Rink amide MBHA resin. All peptides were generated in workable yields (7–12% yield with >95% purity). All vaccine constructs were purified using preparative RP-HPLC in yields similar to lipo-/peptides of similar length previously synthesized [31,33,46]. Broad peaks were observed in the RP-HPLC trace for **VC-1** and **VC-2**. This is associated with the mixed enantiomers of the C16 lipo-amino acid used in their synthesis (Appendix A). 

### 3.2. Physicochemical Evaluation

**VC-1** and **VC-2** self-assembled in water forming small nanoparticles (Table 1). **VC-1** and **VC-2** both contain a lipid core self-adjuvanting moiety (S-S-C_16_-C_16_, Figure 3), which has been shown to enable synthetic vaccine constructs to self-assemble upon dissolving in aqueous solutions (e.g., water or buffer). In this study, **VC-1** and **VC-2** size distribution analysis as measured by DLS showed three distinct peaks (Table 1 and Appendix A) with a PDI of 0.42 for **VC-1** and 0.33 for **VC-2**. A PDI between 0.1 and 0.4 suggests the sample is moderately polydispersed, with a PDI > 0.4 suggesting a highly polydispersed sample. Ideally, a lower PDI is better in vaccine development, showing uniformity in the sample [47]. TEM images of the vaccine constructs showed that the nanoparticles formed worm and rod-like particles (Appendix A). The high PDI generated in this study is associated with self-assembly into rod-like particles, which are 110 nm in diameter for **VC-1**, and around 130 nm in diameter for **VC-2**. The particle size and shape observed is similar to the rod-like and worm-like particles previously reported for self-assembled lipidated constructs [32,48]. Particle aggregations were also observed in the TEM images (Appendix A). The different amino acid composition between **VC-1** and **VC-2** attributed to the different particle sizes formed during self-assembly. 

### 3.3. In Vivo Immunological Study

**VC-1** contains the B cell epitope (J8) identified from the conserved region of the GAS surface M protein. **VC-2** contains the 88/30-epitope from the variable region of the GAS M protein located on the surface of the GAS isotype 88/30 (Figure 1). Previous studies have shown both epitopes to successfully induce a protective immune response in vivo and in vitro [23].

In vivo immunological evaluation was performed in outbred female ARC Swiss mice (Figure 4A). All mice received four intranasal immunizations (single primary immunization and three boosts) on days 0, 21, 42 and 63 under anesthesia with isoflurane. Mice (*n* = 5 mice/group; 7–8 weeks old) were intranasally administered (10 μL/nostril) on days 0, 21, 42, and 63 with vaccine constructs (**VC-1** or **VC-2**; 60 μg in 20 μL sterile Milli-Q water, 10 μL per nostril). Mice of the positive control group were administered the **VC-3** vaccine (60 μg in 20 μL sterile Milli-Q water; 10 μL/nostril) co-administered with commercial CTB adjuvant (10 μg) (**VC-3+CTB**). The negative control group received sterile Milli-Q water (10 μL/nostril). Blood was collected from the tail tip one day prior to each immunization and 14-days following the final boost. Serum was then analyzed by ELISA for total epitope-specific IgG antibodies. 

At the final bleed (day 77; Figure 4A), both **VC-1** and **VC-2** showed significantly higher serum IgG antibody titers when compared with the negative control group (sterile Milli-Q water only, Figure 4B). Mice treated with sterile Milli-Q water did not show any J8- or 88/30-specific IgG responses. Mice treated with both **VC-1** and **VC-3+CTB** produced significantly higher IgG antibody titers against the J8-epitope, and there was no significance between the IgG titers between **VC-1** and **VC-3+CTB** (Figure 4B). All mice treated with **VC-3+CTB** did not produce any IgG against the 88/30-epitope, which was expected since no 88/30-epitope was present in the structure. **VC-2** treated mice produced significantly higher IgG titers against the 88/30-epitope compared to all other groups (Figure 4C) but did not produce any cross-reactive IgG antibodies against the J8-epitope (Figure 4B).

The ELISA results suggest that mice treated with both **VC-1** and **VC-3+CTB** (positive control) can elicit an immune response against J8, showing no significant difference in IgG production titers between both groups (Figure 4B). There was no significant cross-reaction between J8- and 88/30- epitopes in this study suggesting that J8 and 88/30 do not have any similar sequence or structures.

### 3.4. In Vitro Opsonisation Assay

To test the opsonic activity of the vaccines anti-J8 and anti-88/30 antibodies in vitro, we used seven randomly selected GAS isolates which was an innovation to the previous studies i.e., five strains or less [32,33,49,50]. The opsonization performed in this study was on seven clinical isolates of GAS isolated from a local Queensland hospital. These seven isolates have been shown to confer resistance to current antibiotic treatments, with an unknown mechanism of action. We believe the opsonization assay against seven isolates performed in duplicate, in every mouse in each vaccine group, is statistically acceptable to differentiate between our vaccine groups. This leads to more reliable results supporting what is observed in the ELISA assay.

In this study, the sera obtained from the mice immunized with **VC-1** (containing the J8-epitope; Figure 3) were able to elicit significant opsonic activity in six out of the seven GAS clinical isolates (Figure 5). Interestingly, sera obtained from all the five mice immunized with **VC-2** (containing 88/30-epitope) were not able to elicit any significant opsonic activity against the selected GAS clinical isolates when compared to the negative control group (pooled sera from 5 mice from the group immunized with Milli-Q water) (Figure 5). We also noticed that mice with the highest IgG antibody titers from the **VC-1** group showed the highest opsonic activity (Figure 5), which was consistent with the literature [51].

A correlation between the ELISA and opsonization assay results has been reported where the in vitro phagocytosis of *Streptococci* and the corresponding IgG antibody titers showed positive linear correlations [51]. Upon comparison of the opsonic activities of the two highest titered mice from the **VC-1** (J8) and **VC-2** (88/30) groups in this study, it was demonstrated that the immunized sera from the **VC-1** group had a significant bactericidal activity when compared to the **VC-2** group (Figure 6). 

From the opsonization results, it is obvious that the J8-epitope (part of the conservative C-terminus region of the GAS M protein) provides protection against most GAS strains located in Australia and worldwide. The inclusion of C-terminus epitopes into our vaccine is highly necessary in the design of any universal vaccine against GAS. On the other hand, the 88/30-epitope, which is a small sequence derived from the N-terminus region isolated from the Northern Territory (Australia) has been shown to produce a strong immune response only against GAS strains carrying this epitope (or similar epitopes which are rare globally), even in other parts of Australia (e.g., Queensland). This confirms the high variability of the N-terminus region of the GAS M protein where we cannot only rely on this section to design a universal vaccine against GAS in the future. Hence, the addition of 88/30 into a synthetic peptide vaccine in conjugation with the J8 antigen can only make the vaccine better, compared with a J8 alone vaccine. We have proved that our J8-based system is protective against many serovars according to the results. This is important and novel for future investigation of peptide-based GAS vaccine development.

## 4. Conclusions

GAS infection and its associated diseases are global challenges, with no commercial vaccine currently available. In this study, we administrated self-adjuvanting vaccine constructs containing J8- or 88/30-epitopes to mice intranasally and assessed sera IgG antibody against the corresponding epitope following a primary immunization and three boosts. We successfully compared the difference in the opsonization against seven GAS isolates between the lipopeptide-based vaccines containing J8 (developed from a conserved epitope; constructed into **VC-1**) and 88/30 (developed from a variable epitope from clinical isolates from Darwin, Australia; constructed into **VC-2**) GAS M protein epitopes in an opsonization model using seven GAS clinical isolates. We found that the vaccine containing the J8-epitope (**VC-1**) had a significant bactericidal activity when compared to the 88/30-epitopes from **VC-2** in six of seven GAS clinical isolates tested. This suggested that in randomly selected clinical isolates, the J8-epitope provides broader protection. Interestingly, the 88/30 N-terminal epitope, identified from GAS isolates from Darwin (Australia) did not show significant opsonic activity against the seven tested isolates. Hence, we have proved that our J8-based system is protective against many serovars, and peptide vaccines against GAS would benefit from epitopes from both the C-terminal and N-terminal regions of the GAS M protein to ensure broad protection across all GAS isolates. 

## Figures and Tables

**Figure 1 vaccines-08-00210-f001:**
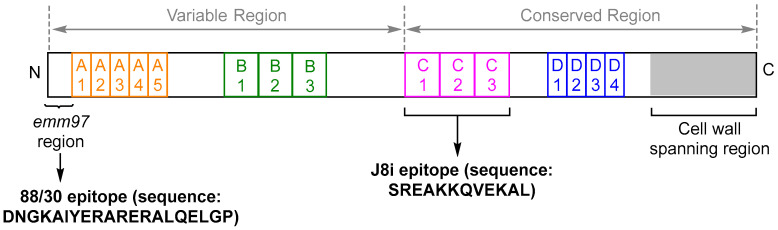
Structure of the GAS M protein [16]. The M protein contains four repeating regions, denoted as A, B, C, and D. The N-terminal of the M protein is variable in sequence with the C-terminal domain being highly conserved. The cell wall spanning region is highlighted in gray. The 88/30-epitope used in this study was identified from the *N*-terminal variable region, and the J8i epitope was identified from the C repeating region [8,17].

**Figure 2 vaccines-08-00210-f002:**
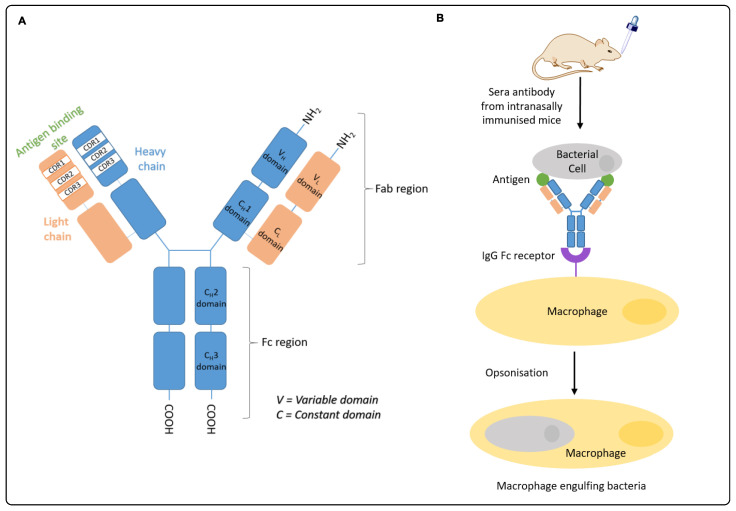
(**A**) Schematic structure of an IgG antibody. IgG have two heavy chains and two light chains. The heavy chain consists of three constant domains, C_H_1, C_H_2, C_H_3 and a single variable domain, V_H_. The Fab region, where the antigen binds, is the complementarity-determining region (CDR) including CDR1, CDR2 and CDR3 on both the heavy and light chains. The Fc region is the primary target for opsonization (in IgG1 and IgG3 only) [27,28]. (**B**) Schematic of opsonic activity of an IgG antibody (generated following intranasal mice immunization in this study). Here, the Fab region of the IgG antibody binds to the bacterial antigen present on the bacterial cell. The IgG Fc region of the IgG antibody then binds to the IgG Fc receptor on the macrophage cell, inducing macrophage phagocytosis of the bacterial cell [29].

**Figure 3 vaccines-08-00210-f003:**
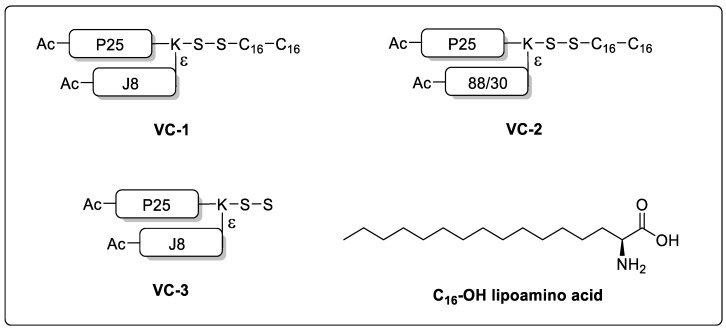
Vaccine constructs evaluated for J8- and 88/30-specific IgG antibodies and their ability to opsonize seven different strains of group A *Streptococci* (GAS). **VC-1** contains the GAS B cell J8-epitope (QAEDKVKQSREAKKQVEKALKQLEDKVQ). **VC-2** contains the *emm6* 88/30-epitope (DNGKAIYERARERALQELGP). **VC-1** and **VC-2** are adjuvanted with a lipid core peptide (LCP) containing two 16-carbon alkyl chains (C_16_-OH lipoamino acids) attached to the C-terminal. The positive control group, **VC-3** is adjuvanted with the commercial mucosal adjuvant, cholera toxin B (CTB). All vaccine constructs contain a universal T helper P25 epitope (KLIPNASLIENCTKAEL).

**Figure 4 vaccines-08-00210-f004:**
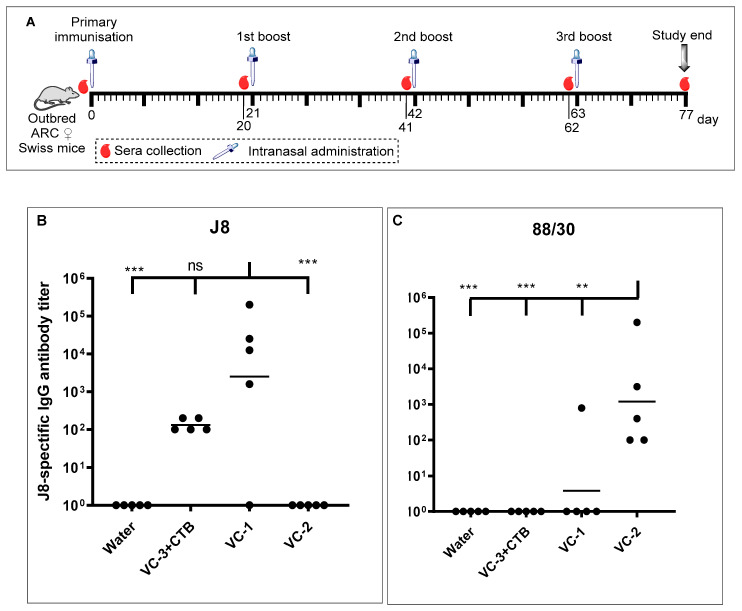
(**A**) Female outbred Swiss mice (6–7 weeks old) immunization schedule. (**B**) J8- and (**C**) 88/30-epitope-specific IgG responses obtained in mice sera on day 77. Geometric mean of epitope-specific IgG titers are represented as a horizontal bar for each group of five mice. Statistical analysis was performed by a one-way ANOVA followed by Tukey’s post hoc test where a probability value of *p* < 0.05 was considered statistically significant (ns, *p* > 0.05; *, *p* < 0.05; **, *p* < 0.01; ***, *p* < 0.001).

**Figure 5 vaccines-08-00210-f005:**
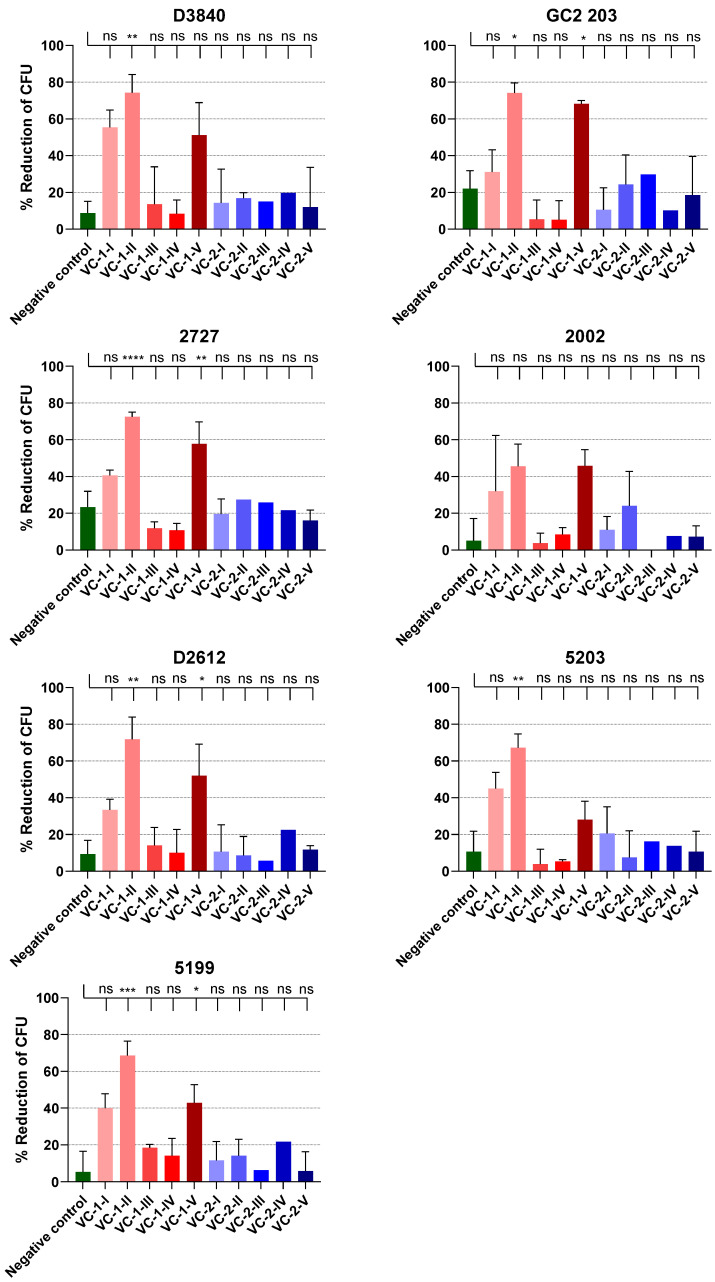
Average opsonization (duplicate from two independent cultures except VC-2-III and VC-2-IV) of sera obtained from mice on day 77 against seven different clinical isolates (D3840, GC2 203, ACM-2727, ACM-2002, D2612, ACM-5203 and ACM-5199) of group A *Streptococcus* (GAS). Negative control pooled sera from five mice in the Milli-Q water group. Results are represented as opsonization percentage compared to reference untreated wells and error is represented as standard error of the mean (SEM). Statistical analysis was performed by a one-way ANOVA followed by Tukey’s post hoc test where a probability value of *p* <0.05 was considered statistically significant (ns, *p* > 0.05; *, *p* < 0.05; **, *p* < 0.01; ***, *p* < 0.001, ****, *p* < 0.0001). CFU = Colony forming unit.

**Figure 6 vaccines-08-00210-f006:**
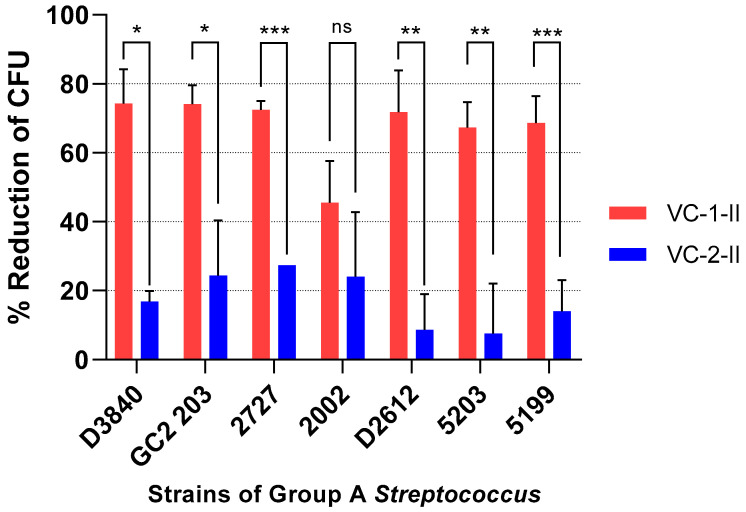
Average opsonization (duplicate from two independent) of sera obtained from mice on day 77 against seven different clinical isolates (D3840, GC2 203, ACM-2727, ACM-2002, D2612, ACM-5203 and ACM-5199) of group A *Streptococcus*. Results are represented as opsonization percentage compared to reference untreated wells and error is represented as standard error of the mean (SEM). Statistical analysis was performed by a one-way ANOVA followed by Tukey’s post hoc test where a probability value of *p* < 0.05 was considered statistically significant (ns, *p* > 0.05; *, *p* < 0.05; **, *p* < 0.01; ***, *p* < 0.001). CFU = Colony forming unit.

**Table 1 vaccines-08-00210-t001:** Size of the vaccine constructs in water.

Vaccine Constructs	Particle Size (nm) ± STD/Ratio (%)	PDI ± STD
**VC-1**	113 ± 26/(88.1) 7 ± 188.1/(11.7) 5560 ± 0/(0.2)	0.42 ± 0.071
**VC-2**	134 ± 42/(91.5) 17 ± 3/(5.8) 4574 ± 847/(2.6)	0.33 ± 0.073

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
