# Peer review of "Opsonic Activity of Conservative Versus Variable Regions of the Group A Streptococcus M Protein"

_vaccines, 2020, doi:10.3390/vaccines8020210_

Round 1
Reviewer 1 Report
The manuscript is a very good work presenting interesting results. Manuscript title describes the article appropriately and lenght of the manuscript is adequate to all content. Description of materials and methods is clear, also results and discussion are coherent. The tables are clear and readable. The references are chosen correctly.
Just some slight corrections are necessary:
- Some medical terms like sepsis, necrotizing fasciitis should be written using italics.
- Please change references formatting - [reference] then "."
- Line 145 - Neisseria meningitidis
- Why only female mice were used in this study?
Author Response
We thank the reviewers dearly for their valuable and insightful comments on our manuscript “Opsonic activity of conservative versus variable regions of the group A Streptococcus M protein (Manuscript ID vaccines-767603)”. We have addressed the recommendations in the manuscript and these changes have been outlined below point-by-point as requested. We hope our responses and comments to the reviewer’s remarks are satisfactory and that the article is now suitable for publication in Vaccines.
With best wishes,
Dr Rachel Stephenson

Reviewer 2 Report
Dai et al. in the "Opsonic...." showed the opsoninic activity of sera obtained as a result of intranasal vaccination with two self-adjuvanting conjugates VC-1, VC-2 and VC-3 adjuvanted with commercial cholera toxin B.
The authors showed that serum after vaccination by VC-1 has better opsonic activity than VC-2. The part about chemical synthesis and characteristic of conjugates is well described and reises no objectives. However, generally the obtained results are not surprising or new. In the literature, it has been well described that conjugates containing the J8 peptides show better opsonic activity or even protection properties.
The work should be improved and some additional experiments nedd to be done. First of all, the manuscript layout must be changed: a very long introduction should be shortened, while the discussion should be developed. Figure 3 should be in the Materials and Methods section. In my opinion, the authors have not sufficiently described why their approach is better than previously known and what new knowledge they bring in GAS fighting/treatment? Is it a nasal administration? - when yes, then they should show the advantages of nasal administration versus intramuscular administration, or maybe is it a selfadjuvanting vaccine? - when yes, then they should show an advantage over other non-selfadjuvanting conjugates. Furthermore, research on opsonization is only initial and "challenge: experiments should be carried out
Minor comments - LPS is recognized by TLR4 (line 138)
Author Response

(The authors gave the same response as above.)

Round 2
Reviewer 2 Report
I appreciate your efforts addressing my concerns but still some changes in manuscript need to be done:
- Figure 3 should be moved to section 2.2 after description of VC-3. Please add some sentences that will lead readers to the Figure 3.
- I appreciate that you tested 7 different GAS strains, but it is an innovation? As I understand innovation - this is a great idea or method, something creative, new imaginations in method executed brilliantly. Innovation is something bigger than only adding two more strains in experiments. Please be so kind and discuss extensively what was the strongest thing in your paper. Please underline the importance of opsonization studies
Author Response
Reviewer 2:
1. Figure 3 should be moved to section 2.2 after description of VC-3. Please add some sentences that will lead readers to the Figure 3.
Authors Response:
Figure 3 have been moved to Section 2.2 following the description of VC-3. We have added sentences to lead the reader to the Figure 3.
2. I appreciate that you tested 7 different GAS strains, but it is an innovation? As I understand innovation - this is a great idea or method, something creative, new imaginations in method executed brilliantly. Innovation is something bigger than only adding two more strains in experiments. Please be so kind and discuss extensively what was the strongest thing in your paper. Please underline the importance of opsonization studies
Authors Response:
In the revised manuscript, we have added more information/discussion to the ‘Introduction’ and ‘Section 3.4 In vitro opsonisation assay’ highlighting the novelty of our research and the importance of the opsonisation studies. Changes can be found on Pages 5, 13 and 15-16. Changes to the ‘Conclusion’ can be found on Page 16.
Page 5: However, there was no direct comparison of the opsonic activity of the C-terminus (J8) and N-terminus (88/30) epitopes in our vaccine delivery system。”
Page 13: “The opsonisation performed in this study was on seven clinical isolates of GAS isolated from a local Queensland hospital. These seven isolates have been shown to confer resistance to current antibiotic treatments, with an unknown mechanism of action. We believe the opsonisation assay against seven isolates performed in duplicate, in every mouse in each vaccine group, is statistically acceptable to differentiate between our vaccine groups. This leads to more reliable results supporting what is observed in the ELISA assay.”
Pages 15-16:“From the opsonisation results, it is obvious that the J8 epitope (part of the conservative C-terminus region of the GAS M protein) provides protection against most GAS strains located in Australia and worldwide. The inclusion of C-terminus epitopes into our vaccine is highly necessary in the design of any universal vaccine against GAS. On the other hand, the 88/30 epitope, which is a small sequence derived from the N-terminus region isolated from the North Territory (Australia) has been shown to produce a strong immune response only against GAS strains carrying this epitope (or similar epitopes which are rare globally), even in other parts of Australia (e.g. Queensland). This confirms the high variability of the N-terminus region of the GAS M protein where we cannot only rely on this section to design a universal vaccine against GAS in the future. Hence, the addition of 88/30 into a synthetic peptide vaccine in conjugation with the J8 antigen can only make the vaccine better, compared with a J8 alone vaccine. We have proved that our J8-based system is protective against many serovars according to the results. This is important and novel for the future investigation of peptide-based GAS vaccine development.”
Page 16:“Hence, we have proved that our J8-based system is protective against many serovars…..”

Round 3
Reviewer 2 Report
I accept the changes made by authors.